# Amnesia after Midazolam and Ketamine Sedation in Children: A Secondary Analysis of a Randomized Controlled Trial

**DOI:** 10.3390/jcm10225430

**Published:** 2021-11-20

**Authors:** Karolline A. Viana, Mônica M. Moterane, Steven M. Green, Keira P. Mason, Luciane R. Costa

**Affiliations:** 1Dentistry Graduate Program, Faculdade de Odontologia, Universidade Federal de Goiás, Goiânia 74000-000, Goiás, Brazil; monicamoterane@discente.ufg.br; 2Department of Emergency Medicine, Loma Linda University, Loma Linda, CA 92354, USA; steve@stevegreenmd.com; 3Department of Anesthesiology, Critical Care and Pain Medicine, Harvard Medical School, Boston Children’s Hospital, Boston, MA 02115, USA; keira.mason@childrens.harvard.edu; 4Department of Oral Health, Faculdade de Odontologia, Universidade Federal de Goiás, Goiânia 74000-000, Goiás, Brazil; lsucasas@ufg.br

**Keywords:** amnesia, memory, dental care, conscious sedation, child, preschool

## Abstract

The incidence of peri-procedural amnesia following procedural sedation in children is unclear and difficult to determine. This study aimed to apply quantitative and qualitative approaches to better understand amnesia following dental sedation of children. After Institutional Review Board Approval, children scheduled for sedation for dental procedures with oral midazolam (OM), oral midazolam and ketamine (OMK), or intranasal midazolam and ketamine (IMK) were recruited for examination of peri-procedural amnesia. Amnesia during the dental session was assessed using a three-stage method, using identification of pictures and an animal toy. On the day following the sedation, primary caregivers answered two questions about their children’s memory. One week later, the children received a semi-structured interview. Behavior and level of sedation during the dental session were recorded. Quantitative data were analyzed using descriptive statistics and comparison tests. Qualitative data were analyzed using content analysis. Triangulation was used. Thirty-five children (age: 36 to 76 months) participated in the quantitative analysis. Most children showed amnesia for the dental procedure (82.9%, *n* = 29/35) and remembered receiving the sedation (82.1%, *n* = 23/28 for oral administration; 59.3%, *n* = 16/27 for intranasal administration). The occurrence of amnesia for the dental procedure was slightly higher in the oral midazolam group compared with the other groups (44.8%, *n* = 13/29 for OM, 13.8%, *n* = 4/29 for OMK, and 41.4%, *n* = 12/29 for IMK). Twenty-eight children participated in the qualitative approach. The major theme identified was that some children could remember their procedures in detail. We conclude that peri-procedural amnesia of the dental procedure was common following sedation.

## 1. Introduction

Sedation is often necessary in pediatric dentistry for children with anxiety, fear, and behavioral management problems [1]. Amnesia, defined as an inability to recall information consciously [2], is a desirable feature of sedation. Most children, as well as their parents, request and expect to avoid such recall, and its presence may adversely impact the future cooperation of children and their families with sedation and dental care [3]. Complete amnesia may be undesirable in certain situations [4] and is also not necessarily associated with poor outcomes, particularly for non-invasive procedures.

The incidence of recall during sedation varies widely by study, with reports from 0% [5] to 100% [6,7,8]. This difference can be attributed to the drug used; in the former study, children were sedated with nitrous oxide and melatonin, whereas in the other investigations, they were sedated with amnestic drugs (propofol, ketamine, or benzodiazepines). Ketamine, benzodiazepines, barbiturates, chloral hydrate, antihistamines, narcotics, nitrous oxide, dexmedetomidine, Propofol, and melatonin have been administered for dental sedation, with limited research for most regarding the incidence of amnesia [9]. Factors associated with amnesia include patient characteristics, level of sedation, and the method used to evaluate amnesia [10]. Establishing memory and recall in the pediatric population is challenging, with quantitative methods most frequently employed [9,11,12].

Most studies of sedation-associated amnesia in children have used quantitative methods to identify the memory of photos, toys, or situations presented during the procedure, as assessed by either parents or children [9]. Such methods lead to heterogeneous results that limit the quality of evidence [9], and there are few relevant qualitative studies.

Qualitative research has the advantages of a more holistic understanding of recall, and real feedback via in-person interviews [13,14]. Furthermore, in clinical trials of procedural sedation, it is pivotal to include other aspects beyond efficacy, efficiency, and safety, such as patient and family-centered outcomes, which includes recall [4]. Accordingly, this exploratory secondary analysis of a randomized controlled trial (RCT) [15] applied quantitative and qualitative approaches to investigate the incidence of amnesia following dental sedation of children. Our hypothesis was that almost all children would show complete amnesia.

## 2. Materials and Methods

### 2.1. Study Design and Ethical Aspects

This secondary analysis of a RCT is a sequential mixed-methods study, combining a complementary quantitative and qualitative approach to enhance the interpretation of the findings [16,17]. The primary aim of the RCT was to compare the efficacy of three sedative regimes on behavior (NCT02447289): IMK group—intranasal midazolam 0.2 mg/kg (maximum dose 5.0 mg, Dormire^®^ injectable solution, Cristália, São Paulo, Brazil) and ketamine 4.0 mg/kg (maximum 100.0 mg, Ketamin S^®^ injectable solution, Cristália, São Paulo, Brazil) [18]; OMK group—oral midazolam 0.5 mg/kg (maximum 20.0 mg, Dormire^®^ oral solution, Cristália, São Paulo, Brazil) and ketamine 4.0 mg/kg (maximum 100.0 mg, Ketamin S^®^ injectable solution, Cristália, São Paulo, Brazil) [19,20]; and OM group—oral midazolam 1.0 mg/kg (maximum 20.0 mg, Dormire^®^ oral solution, Cristália, São Paulo, Brazil) [20]. The midazolam doses followed prior recommendations [18,19,20] based on 78% and 36% bioavailability for the nasal and oral routes, respectively [21,22].

In the RCT, the child received one dental restoration under local anesthesia with 1:100,000 epinephrine (Alphacaine 2%, Nova DFL, Rio de Janeiro, Brazil) and rubber dam isolation performed by one of four certified pediatric dentists. The primary caregivers were seated with the child in the dental chair throughout the procedure, and the treatment was filmed. Sedation was administered in accordance with American Academy of Pediatrics and the American Academy of Pediatric Dentistry guidelines [23], with continuous physiological monitoring including depth of sedation, heart rate, respiratory rate, and oxygen saturation. The dental team, interviewers, children/primary caregivers, and data analyst were blinded to the group assignment. The RCT protocol [24] and primary results (children’s behavior during dental sedation) [15] have been published separately previously. The sample size estimation and randomization were based on the primary aim of the RCT (behavior). Although ‘memory’ was pre-planned as a secondary outcome (2015), we altered the original protocol when we found that children less than 3 years could not complete the memory tests owing to their cognitive development. Indeed, a systematic review [9] found that, although there is no gold standard to assess children’s memory of procedural sedation, the three-stage method [2] is predominantly used with children over 3 years old, supporting our approach. Therefore, we excluded the missing cases for the ‘memory’ analysis in this secondary analysis.

This study was approved by the Research Ethics Committee of the Universidade Federal de Goiás (CAAE 36411214.1.0000.5083) and followed the ethical principles stated in the Declaration of Helsinki [25]. All parents/guardians voluntarily consented to their and their children’s participation.

### 2.2. Participants

This secondary analysis included 35 of the 84 children from the RCT. For the present evaluation of amnesia, we limited assessment only to those children > 3 years old, as the ability to assess memory in toddlers is beyond the scope of our quantitative and qualitative data methodology. Participants were children three to six years old, as well as their respective primary caregivers (usually parents). All children were ASA I or II [26], had no identifiable risk factors for airway obstruction [27], had no neurological or cognitive impairment, had early childhood caries, and had demonstrated negative behavior [28] during a previous dental visit.

Our sample size of 35 children for this quantitative analysis was based upon the subset of all children 3 to 6 years of age enrolled in the primary trial. For the qualitative analysis, the sample size was estimated to be sufficient according to the principle of theme saturation, i.e., adding more participants would not add new information [29].

### 2.3. Proceedings of the Randomized Clinical Trial Phase

To ensure the blinding during the RCT, a physician delivered the sedatives following the pre-determined sequence: first (T0), the oral syrup was administered; ten minutes later (T10), an intranasal solution was administered; then, thirteen minutes after the oral syrup (T13), another intranasal solution was delivered. Twenty minutes after the oral syrup (T20), the dental treatment started. According to this sequence, the IMK group received a placebo syrup at T0, intranasal ketamine at T10, and intranasal midazolam at T13; the OMK group received midazolam and ketamine orally at T0 and intranasal placebo at T10 and at T13; and the OM group received oral midazolam at T0 and intranasal placebo at T10 and at T13.

### 2.4. Procedures to Assess Amnesia

We developed an instrument to assess amnesia through visual recognition and recall tests, based on the three-stage method (encoding, retention interval, and test phase) [2] (Figure 1). The instrument was pre-tested in 27 non-sedated children (14 girls, 51.9%), aged 42–82 months (mean: 60.8; SD: 11.1), which established it as adequate for this purpose. In this instrument pre-test, the pictures and the animal’s evocation rates were above 85% and the picture and animal recognition was above 77%.

In the encoding phase, children were exposed to two pictures and one animal toy. Fourteen minutes after administering the oral syrup, one standard picture (bicycle) was shown to the child for 5 s on a tablet. The patient was asked to identify the image and name it. If the child was unable to do so, the researcher spoke the name and asked the child to repeat it. If the child did not talk, the researcher reminded them to look at the picture. These procedures were repeated nineteen minutes after the oral syrup with another standard image (heart). Finally, these procedures were repeated immediately before the dental prophylaxis, when the dentist showed an animal toy (frog) to the child. The choice of these times for presenting the stimuli was based upon prior research [30].

The retention interval corresponds to the gap between the encoding and test phases. The test phase was performed when children met the sedation discharge criteria, i.e., awareness similar to pre-sedation or close to normal; satisfactory, stable cardiovascular function and airway adequacy; easy arousability with protective reflexes intact; the ability to speak and sit with minimal assistance (if applicable); and adequate hydration status [23]. The researcher then asked the children if they could recall the pictures/animal toy (recall test). In the end, patients were asked to identify the pictures shown previously among four images—two target (old) and two distractors (new) (recognition test). This procedure was then repeated with the animal toy.

Primary caregivers were advised not to speak about the procedures with their children. The following day, the researcher called the children’s primary caregivers with two questions: (1) Did the child say something regarding the performed interventions? If yes, what? (2) Do you think the child remembers the performed interventions? Why?

The qualitative data collection took place, aiming to confirm or not the findings of children’s amnesia obtained through the three-stage method. One week after sedation, two trained dentists, supported by a psychologist, conducted a semi-structured interview with the children, using a guide and the literary book “Peppa Pig goes to the dentist” to catch children’s answers. The child was asked about memories about the dental appointment on each page, such as “Did you drink any syrup at the dentist?” The interviews were video recorded and later transcribed verbatim by the two researchers. This instrument was pre-tested with three children and, when confirmed to be viable for the objectives of this study, these cases were included in the final analysis.

### 2.5. Procedures to Assess Children’s Behavior and Level of Sedation

Each child’s behavior was assessed using the Frankl scale [28] by the treating dentist at the end of the appointment: (1) definitely negative behavior—refuses dental treatment; (2) negative behavior—is reluctant to accept the dental treatment; (3) positive behavior—agrees with the dental treatment; and (4) definitely positive behavior—shows good behavior, interest in the dental procedures, and has fun with the situation. These dentists were trained and calibrated to the scale by watching pediatric dental treatment session videos. The Kappa values for inter-examiner agreement for the four dentists varied from 0.63 to 0.86, depending on the Frankl scale score, whereas the Kappa for intra-examiner agreement was 1.

To assess the level of sedation, one pediatric dentist with experience in dental sedation watched the videos obtained during the treatment and classified the sedation as minimal (relaxed and awake), moderate (relaxed and drowsy), or deep (drowsy to lightly sleeping) [31]. The same dentist evaluated these same videos one month later, with a Kappa for this intra-examiner agreement of 0.93.

### 2.6. Data Analysis

Data were analyzed using quantitative and qualitative approaches. For the quantitative analysis, we assessed recall and recognition of the pictures and animal’s toy, recollection of sedation administration, and the yes/no responses by primary caregivers. For the qualitative analysis, we considered the open question answered by primary caregivers and the interviews with children. The integration of the two sources took place according to the convergent triangulation model: the researcher collects and analyzes the quantitative and qualitative data separately and then merges the different results to compare or combine them [13].

### 2.7. Quantitative Approach

Amnesia, considered in this study as an inability to recall information consciously [2], was quantified as follows:(1)Lack recall of each picture and the animal toy.(2)Lack of recognition of pictures, including both the hit rate (correct recognition of target pictures) and false alarm rate (incorrect identification of distractor item as a target picture). A difference between the hit and false alarm rate equal to one was considered recognition.(3)Lack of recognition of animal toy: measured in a similar way of recognition of pictures.(4)Amnesia of dental procedure: an absence of recall or recognition of animal toy because it was the stimuli shown during the appointment.(5)Primary caregivers’ response to the question: “Do you think your son/daughter remembers the performed interventions?”(6)Amnesia of oral and intranasal administration: taken from interviews.

An exploratory analysis was performed to compare these variables regarding sedatives’ groups, level of sedation, and children’s behavior, which was dichotomized in positive or negative behavior (Frankl scale).

Data were analyzed using descriptive statistics and comparison tests using the statistical software IBM SPSS 24.0 (IBM Corporation, Armonk, NY, USA), with a significance level established at 5%. The Cochran’s Q test followed by McNemar was used to compare the frequency of amnesia of picture 1, picture 2, and animal toy in the total sample and each group of sedatives. The Kappa test was used to check the agreement between children and primary caregivers, and the strength of agreement was considered according to the following values: poor (<0.00), slight (from 0.00 to 0.20), fair (from 0.21 to 0.40), moderate (from 0.41 to 0.60), substantial (from 0.61 to 0.80), and almost perfect (from 0.81 to 1.00) [32].

### 2.8. Qualitative Approach

Two independent reviewers analyzed the transcripts using qualitative content analysis. First, transcripts were explored exhaustively to obtain a general sense. Codes were then independently developed based on text excerpts using the software NVivo 11 (QSR International Pty Ltd., Melbourne, Australia), with disagreements resolved by consensus. Finally, the codes were sorted into categories, and a theme was developed [33]. There were no aprioristic categories, and the responses were analyzed using the inductive method.

## 3. Results

Of the 84 children included in the trial, 49 were excluded from the present study (55.1% girls, mean age 35.1 months, standard deviation (SD) 10.1, 55.1% negative behavior): 44 because of the age being under 36 months, 4 were sleeping during the procedure, and 1 refused to respond. A total of 35 children, 19 girls (54.3%), aged between 36 and 76 months (mean 52.4 months, SD 11.8), and their primary caregivers participated in the quantitative analysis. Among them, seven children/primary caregivers were unable to be interviewed at one week follow-up and were excluded from the qualitative analysis: two were lost to follow-up despite three attempts, two did not understand the instructions, and three refused to participate (Figure 2). Data saturation for the qualitative analysis was achieved with 28 interviewed participants (Appendix A) (mean age 55.1 months, SD 11.2; 64.3% female) and allowed the content analysis.

### 3.1. Quantitative Analysis

The mean duration of the procedure was 25.0 min (SD 8.8). The mean retention interval (the difference between the encoding and test phases) was 92.0 min (SD 26.0). Many children did not recall the pictures or the animal toy and did not recognize them (Figure 3).

Although most children showed amnesia for the procedure (82.9%, *n* = 29/35), the primary caregivers were not as consistent in their confirmation of amnesia (52.9%, *n* = 18/34), with only fair concordance with the children’s report (Kappa = 0.325). There was a difference in recall of drug administration with respect to the nasal versus oral route of administration (82.1%%, vs. 59.3%, respectively). Positive behavior (51.4%, *n* = 18/35) and a moderate sedation level (68.6%, *n* = 24/35) predominated among all children.

Amnesia was higher with moderate (versus minimal) sedation, oral midazolam alone (versus ketamine and midazolam), and those who display negative behavior (Table 1); statistical tests were not performed because of the subgroups’ small size.

When considering the different sedative regimes, groups were similar regarding sex and age. There was a predominance of negative behavior in the IKM and OM groups. Regarding the sedation level, there was a predominance of moderate sedation in all groups. No child exhibited deep sedation (Table 2). Most children did not recall or recognize pictures/animal toy, except when considering children that received IMK; in this group, few children did not recall picture 1 (Figure 3).

In the intragroup analysis, children who received IMK had the least amnesia of picture 1, compared with picture 2 and the animal toy (Cochran’s Q test, *p* = 0.001) (Figure 4). Similar differences were not observed when the OM (*p* = 0.165) and OMK (*p* = 0.368) groups were analyzed.

### 3.2. Qualitative Approach

The central theme identified was that some children could remember in detail the procedures performed while they were sedated. This theme was composed of four main categories: the child clearly remembered; the child did not remember; the child had uncertain remembrance; the child had abilities to remember (according to primary caregivers) (Table 3).

### 3.3. Category: The Child Clearly Remembered

Children recalled the dental procedures: “my tooth was treated” (C1, female, 61 months) (referring to what occurred after she drank the sedatives), and mentioned the animal toy that was shown to her during the dental treatment. “I put the frog to sleep” (C2, male, 73 months). Other children spoke about treatment for their parents: “She said that she had her teeth treated, but she did not put the orthodontic appliance on them” (C3, female, 60 months); “She said that dentist put a ‘pool’ in her tooth” (C4, female, 52 months) (referring to the rubber dam isolation). Some children reported their sensations during the procedure at home: “She said she rolled over and was dizzy. She also said it did not hurt” (C7, female, 43 months); “She just said that she took a little medicine and that she got dizzy. Then she sang a song all the time, that my mother, who was with her, said that the doctor sang to her” (C8, female, 69 months).

### 3.4. Category: The Child Did Not Remember

Some children insisted that they did not go to the dentist: “He just looked at it, and Mom and I went away” (C9, female, 45 months).

### 3.5. Category: The Child Had Uncertain Remembrance

Others were inconsistent in their answers: they stated that they have not had their teeth treated: “I just took medicine and left” (C10, male, 50 months) and after said that they had undergone dental treatment: “the dentist took care of this tooth here” (child pointed to the tooth) (C10, male, 50 months).

### 3.6. Category: The Child Had Abilities to Remember

Regarding primary caregivers’ perception of amnesia of children, some of them considered that their child had the personal abilities to remember: “he is clever” (C11, male, 76 months) and “I think she remembers because she barely forgets things” (C12, female, 62 months), whereas others thought that their child’s state during the treatment could lead them to remember: “he was not so calm with the sedation” (C13, male, 45 months).

After triangulation, convergence was found between the quantitative and qualitative approaches in remembering or not remembering the procedures in thirteen children: six of them recalled in both analyses, and seven children showed amnesia. Among the children that showed divergent results (*n* = 14), eleven remembered the procedure according to the quantitative approach, but showed uncertain remembrance during the interview (*n* = 7) or seemed not to remember during the interview (*n* = 4).

## 4. Discussion

This study found that many children undergoing dental sedation showed amnesia for the procedure. However, others remembered in detail the perioperative events beginning from the administration of sedative to the end of the procedure, including the presence of discomfort and dizziness. We expected that almost all children would show complete amnesia, as midazolam and ketamine should each theoretically confer amnesia. Benzodiazepines and ketamine can impair memory by decreasing attention and arousal and interfere directly with the memory process [10]. However, some of our children fully described the dental procedures in detail while sedated, despite the sedation level or the sedative group. This recall and memory of unpleasant experiences is clinically relevant, as a negative experience can precipitate avoidant behaviors with aversive clinical reactions [34] and dental phobia for future dental appointments [35].

Our results associate the depth of sedation with recall, with minimal sedation manifesting less amnesia than moderate. Per definition, patient responsiveness is higher with minimal sedation, possibly a factor that lends itself to the suggestion that, at lower depths of sedation, the greater the attention and response to stimulus, the higher the subsequent risk of recall [36]. These findings support similar findings with dexmedetomidine and propofol, supporting that children who are verbally responsive to visual and verbal stimuli have a higher incidence of recall [11,12]. It should be noted, however, that increasing the level of sedation increases the risk of respiratory and airway adverse events [23].

A slightly higher occurrence of amnesia was found among children sedated with oral midazolam compared with those sedated with midazolam and ketamine. It could be argued that this may be due to the higher dose given in this group compared with the others. Nonetheless, the evidence is weak regarding dose-dependent amnesia with benzodiazepines [9]. Even low doses of benzodiazepines can cause amnesia. Thus, increasing the dose would not necessarily increase the frequency of amnesia, as we note in our study.

Although many children in our study recalled receiving the oral sedative, fewer did not recall/recognize the pictures and the animal toy shown to them 14 to 23 min after oral administration, respectively. Intranasal midazolam and ketamine demonstrated higher rates of amnesia for picture 2 and the animal toy compared with picture 1. This finding is probably due to the moment of the picture 1 encoding (T14—four minutes after ketamine and one minute after midazolam administration). Considering that the onset of action of ketamine is around 5 min [37] and that of midazolam is 10 min [38], it is possible that the child was not yet under the sedative’s effect. Moreover, neuroimaging studies have reported that midazolam and ketamine differently affect brain functional connectivity related to children’s cognitive abilities while sedated [39,40]. Post-encoding stress can also impact functional connectivity and memory performance [41]. Combining these results, we could hypothesize that children could feel more distressed because of their altered level of consciousness. Therefore, it is advisable for children and parents to avoid unduly dwelling on past painful and distressing events [42].

Sedation outcomes should include patient-centered and clinician-centered measurements [4]. Interviewing younger children is a challenge owing to their limited cognitive and linguistic ability and their tendency to fabricate memories and experiences. Nevertheless, different methodological approaches can be used to elicit their experiences [43]. In this study, we attempted to carry out the interview using diverse strategies to ensure accurate and credible information, such as performing the late interview and using prompts/props and open questions to encourage children to speak. A late interview is a valuable approach in detecting memory because children are no longer under the drug effect and no longer in a stressful clinical environment [44]. A challenge to late interviews is that some children may have forgotten the treatment received or are in doubt as to their treatment.

Our study has several limitations. Given the limited sample for the different subgroups, this must be considered an exploratory study that is intended to be descriptive rather than hypothesis testing. Further, as a sequential mixed-method study, its quantitative and qualitative approaches should be considered together. An additional limitation is that it is possible that the primary caregivers may have influenced their child’s responses, although we emphasized to them the importance of not speaking about the procedure with children at home, and we performed the interview without the presence of primary caregivers. To corroborate the child’s answers, we also questioned primary caregivers [43], with fair agreement noted, as has been previously observed [45]. This discord could represent a lack of parental perception about their children or just a difference in perspective [45]. We did not study tactile, taste, or olfactory memory, which might differ from visual. Finally, while noting the weak evidence regarding dose-dependent amnestic properties, the potential impact on amnesia of different drugs’ doses is uncertain.

This study is exploratory, so our findings should be seen as the first step in understanding children’s procedural sedation memory. The sample size is a limitation of the secondary analysis, that is, our sample size was dependent on the number of eligible subjects taken from the primary analysis, thus some of our contrasts may have been underpowered. We did not perform subset analyses based upon differing drugs and doses because these were not part of our planned study objectives, and because such contrasts would likely be underpowered. Finally, our observed loss to follow-up in the qualitative analysis may bias its result and its interpretation, although data saturation was achieved with the remaining participants. Future studies would benefit from larger sample sizes. The postoperative behavior of children that remember and do not remember the procedure can also be evaluated in future studies.

Our study is important because it demonstrates that a qualitative assessment of sedation, evaluating both the child and the primary caregivers, is a feasible means of assessing patient recall and amnesia during pediatric dental sedation. Importantly, the depth of sedation may predict the risk of recall and, despite achieving adequate depths of sedation for success of the procedure, there is still the risk of patient recall and subsequent negative behaviors and phobia around future dental procedures.

## 5. Conclusions

Peri-procedural amnesia of the dental procedure is common following sedation; however, a few children can remember the procedures in detail. Sedation providers must account for the possibility of recall and risk of future procedure and sedation-related phobias when administering dental sedation to children in the office-based setting.

## Figures and Tables

**Figure 1 jcm-10-05430-f001:**
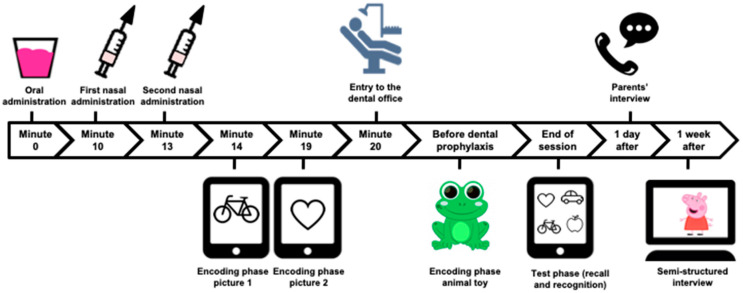
Steps of the amnesia assessment.

**Figure 2 jcm-10-05430-f002:**
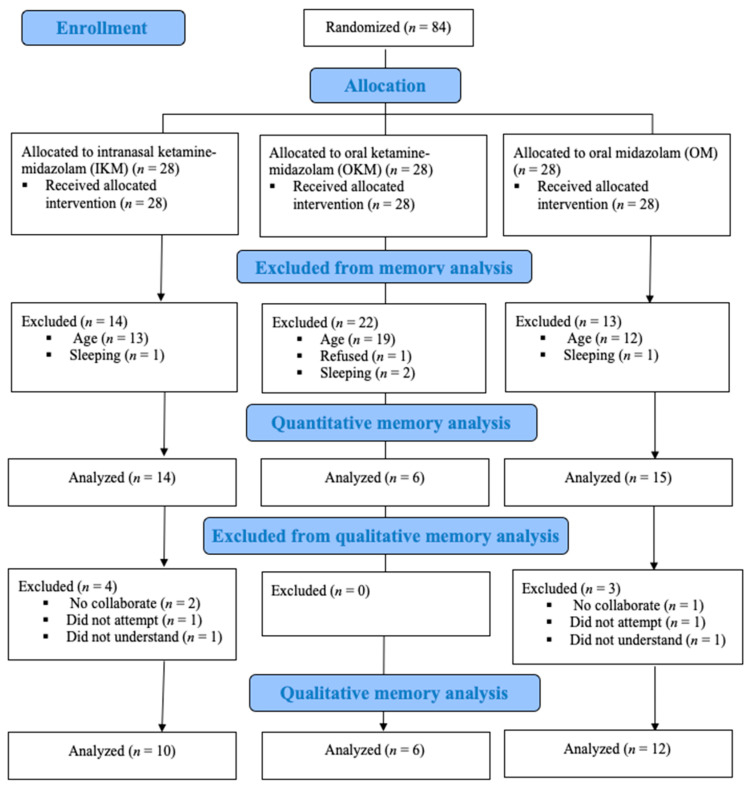
Flow diagram of study progress stages.

**Figure 3 jcm-10-05430-f003:**
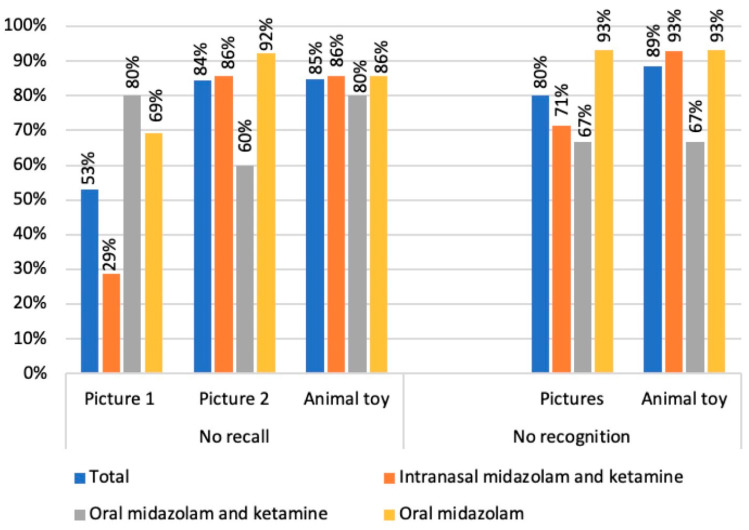
Occurrence of no recall and no recognition of pictures/animal toy among all the participants, and according to sedative groups.

**Figure 4 jcm-10-05430-f004:**
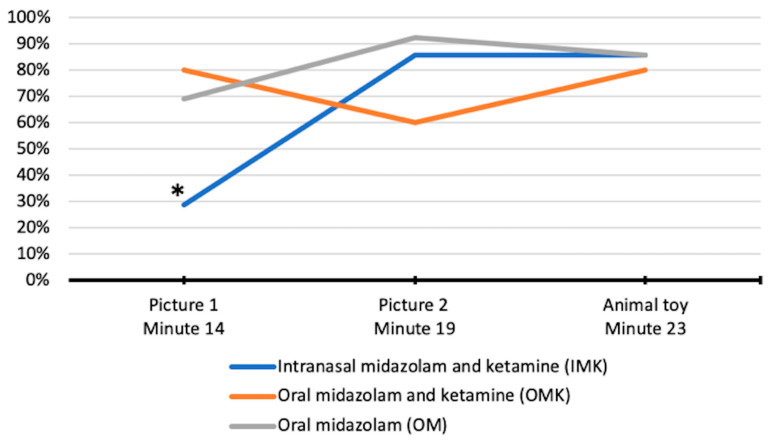
Comparison of the percentage of children who did not recall the pictures and the animal toy in each sedative group. * Differences between amnesia of picture 1 and animal toy (McNemar *p* = 0.003) and amnesia of picture 1 and picture 2 (McNemar *p* = 0.006) in children who received the intranasal midazolam and ketamine.

**Table 1 jcm-10-05430-t001:** Frequencies of amnesia during pediatric procedural sedation according to different perspectives and variables.

Variables	Amnesia of the Dental Procedure (Absence of Toy Recognition or Recall) (*n* = 35)		Amnesia According to Primary Caregivers’ Report (*n* = 34)	
	Yes	No	Yes	No
Sedation level				
Minimal	7 (24.1%)	4 (66.7%)	5 (27.8%)	6 (37.5%)
Moderate	22 (75.9%)	2 (33.3%)	13 (72.2%)	10 (62.5%)
Sedative groups				
Intranasal midazolam and ketamine	12 (41.4%)	2 (33.3%)	8 (44.4%)	5 (31.3%)
Oral midazolam and ketamine	4 (13.8%)	2 (33.3%)	1 (5.6%)	5 (31.3%)
Oral midazolam	13 (44.8%)	2 (33.3%)	9 (50.0%)	6 (37.5%)
Children’s behavior during sedation				
Negative	15 (51.7%)	2 (33.3%)	10 (55.6%)	6 (37.5%)
Positive	14 (48.3%)	4 (66.7%)	8 (44.4%)	10 (62.5%)
Total	29 (82.9%)	6 (17.1%)	18 (52.9%)	16 (47.1%)

**Table 2 jcm-10-05430-t002:** Characteristics of children according to sedative groups.

**Variables**	**Groups**		
	Intranasal Ketamine/Midazolam (*n* = 14)	Oral Ketamine/Midazolam (*n* = 6)	Oral Midazolam (*n* = 15)
Sex, *n* (%)			
Female	7 (50.0%)	3 (50.0%)	6 (40.0%)
Male	7 (50.0%)	3 (50.0%)	9 (60.0%)
Age in months, mean (SD)	54.6 (12.8)	57.3 (11.2)	48.6 (11.0)
Behavior during the dental session, *n* (%)			
Positive	6 (42.9%)	5 (83.3%)	7 (46.7%)
Negative	8 (57.1%)	1 (16.7%)	8 (53.3.%)
Level of sedation			
Minimal	3 (21.4%)	2 (33.3%)	6 (40.0%)
Moderate	11 (78.6%)	4 (66.7%)	9 (60.0%)

**Table 3 jcm-10-05430-t003:** Content analysis outcomes related to children’s amnesia.

Theme	Categories	Codes–*Children…*
Some children can remember in detail procedures performed during dental sedation	Clearly remembered	recalled the sedative administration
		recalled the animal toy
		recalled specific dental procedures or the whole session
		talked about dental treatment/memory assessment at home
		reported sensations during the procedure (pain, dizziness)
	Did not remember	forgot the animal toy
		did not remember specific procedures or the whole session
	Uncertain remembrance	gave some clues of remembering the treatment during the interview, but they were inconsistent
		recalled episodes that could have happened during the clinical examination or in the recovery (e.g., slept at the dentist, taken photo with the dentist, sang the same song that dentist sang)
	Abilities to remember (according to primary caregivers)	are clever
		have their own opinion
		are agitated

## Data Availability

The data may be made available by the corresponding author through email.

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
