# Peer review of "Amnesia after Midazolam and Ketamine Sedation in Children: A Secondary Analysis of a Randomized Controlled Trial"

_jcm, 2021, doi:10.3390/jcm10225430_

Round 1
Reviewer 1 Report
This manuscript describes a secondary analysis of a previously published randomized trial of efficacy of three sedative regimens. The study determined the levels of amnesia following sedation. Sample sizes were small and statistical comparisons among the 3 sedative regimens were not performed.
The manuscript would benefit from additional editing for word choice and grammar.
Terminology between “parents: and “primary caregivers” should be consistent.
Figure 4: The asterisk in the figure should be defined in the legend.
Reviewer 2 Report
I would like to thank the authors for their manuscript titled "Amnesia after midazolam and ketamine sedation in children: a secondary analysis of a randomized controlled trial". This is an interesting approach to a secondary analysis of the primary randomized controlled trial. Please see my comments below.
Introduction
- Page 1, line 40: The first sentence is unclear to me. I believe the authors meant to say "...for the management of anxiety and fear in cognitively and behaviorally challenged children undergoing dental procedures"
- Page 1, line 43: commas are missing. "Most children, as well as their parents, request..."
- Page 2, line 51: Propofol is used in dental sedation and should be included in this list
- There is a lack of primary goal of this project described. The final sentence of the introduction says you did secondary analysis of the RCT in order to quantitatively and qualitatively better understand the incidence of amnesia. Is there a hypothesis associated with this goal. In the conclusion you state this is an exploratory analysis. Please mention a hypothesis (if one existed) along with the goal of the study being an exploratory analysis (if that was the case).
Materials and methods
- Page 4, line 146: "her" should be "them"
- Page 4 line 161: Were there any steps taken to prevent parental disruption of the study other than asking them to "not speak about the procedures with their children". Was this confirmed on subsequent follow ups? If so, was this recorded or were subjects removed if parents did talk about the procedure with the children?
- Page 5, line 173: Was the instrument described tested thoroughly enough? Is pre-testing with 3 children enough to prove it is viable? Please support this or describe your decision making.
Results:
- Figure 3: Is this for all 3 groups combined? Would this be more beneficial to split into all 3 individual groups.
- Table 1: The variable column has subheadings (Sedation level, Sedative groups, and Children's behavior during sedation). These should be underlined or bolded to differentiate between the sub categories such as minimal or moderate.
Discussion
- Overall, the statements in the discussion should be lightened. The authors do state that their sample size likely was underpowered but this was a limitation of secondary analysis (but this is not mentioned until the final page of the paper). Given the final sample sizes of 10, 6, and 12, it is difficult to make strong statements supporting these conclusions.
- Page 11, line 323: do you mean all groups would provide complete amnesia or only the mixture of ketamine /midazolam? If the latter, please clarify if you thought the mixture of ketamine/midazolam would offer complete amnesia for the oral or intranasal or both. If this is your primary hypothesis, please include it in the goals of the study earlier in the manuscript.
- Page 11, line 342: The sentence "Thus increasing the dose of the sedative would not increase the occurrence of amnesia" Based on the results showing the oral midazolam (at a higher dose) showed was associated with high amnesia, you could mention an exception to this statement.
- Page 12, line 371: See above comment regarding identifying children who were interfered with by their parental discussion. Was this measured?
- If the association between increased sedation and decreased recall/increased amnesia is made, it should be worth noting that increased sedation is not without risk (increased incidence of airway complications, hypotension, etc)
Reviewer 3 Report
This is a well written and design randomized controlled trial (RCT) addressing an important area –sedation and amnesia in pediatric anesthesiology. In addition, this secondary analysis can offer valuable information about current practice.
I have some minor comments which need to be modified:
Page 6, Line 238 “mean age 35.1” in Results section may be confusing. You should give an unit (e.g., months) for readers to understand easily.
Page 12,
In one of limitations, this secondary analysis of RCTs lost to follow-up 4/14 (28.6%) in IKM group and 4/15 (26.7%) in OM for qualitative memory analysis. This large lost to follow-up can compromise the result and its interpretation. Therefore, the authors should state it in the limitations.
Author Response
Please see the attachment.

This manuscript is a resubmission of an earlier submission. The following is a list of the peer review reports and author responses from that submission.
Round 1
Reviewer 1 Report
Line 41 "may have to have" I think should read "may have"
Line 46 - 47 Incidence . . . "have" I think should be "has"
Line 52-56 needs rewriting, it is unclear what the authors are trying to say
randomized clinical trial - how was it randomized? It is completely unclear to me whether the children got oral and intranasal medication and when saline was given as placebo. Please be more specific in identifying the treatment groups adn whether the groups were equal in age as developmentally there is an enormous different between a 3 year old and a 6 year old.
All children seemed to get midazolam in some form so it is unclear how line 291 is relevant
Was the recall instrument more reliable in the older patients?
It is unclear how the qualitative results "inform" the quantitative results
LIne 291 - "Our results unanticipated?" not sure what this means
Line 310 Why would a shorter onset of action produce more amnesia at a later time period? What is the duration of effect?
Please address newer science that may suggest that lack of recall is distressing when the patient is not expecting general anesthesia. Recall of procedures occurring that are in an non-stressful or distressing environment may actually be better than complete amnesia though obviously the developmental stage of children makes that difficult to study
Reviewer 2 Report
Well done study of a difficult to assess parameter. Amnesia in the oral midazolam alone group was higher as reported. This may be due to the higher dose given in this group compared to the others. Perhaps a comment concerning dosing as it might relate to amnesia would be appropriate.
You reported that the amnesia in the group receiving only oral midazolam was more pronounced than in the other groupings. Please comment on why this may be so. I think dosing had more to do with this than the route of administration, as this group received the highest dose of midazolam, which has demonstrated dose-dependent amnestic properties. Also, please comment on why different dosages of midazolam were used in the various groups. Dosing should have been consistent across all groups. Lack of recall is difficult to assess in pediatric patients. Your study model was designed well enough to assess this, but the dosage differential among routes of administration could have affected the results. Please include comments regarding the dosing differential when you discuss the study’s possible limitations.Author Response
Please see the attachment

Reviewer 3 Report
The incidence of amnesia following these sedatives have not been studied in depth.There are a number of problems in understanding the administration of children's medications. Were the children also treated with nitrous at the time of treatment? The result of sedation depth is unclear.Is it possible to get a references for the dosages of the drugs?I would like to ask - the number of children tested was 28 while the table has 29.The subject is interesting and important, but the number of children is small and there is no focus on methods and materials. The diagram is beautiful.
Round 2
Reviewer 1 Report
While the presentation of the methods has improved greatly, it is still unclear to me what this study adds. There are many studies of midazolam administration in adults that look at recall and no amount of midazolam is associated with 100% lack of recall. Recall is also not necessarily associated with poor outcomes particularly for non-invasive procedures.